# A Self-Powered DSSH Circuit with MOSFET Threshold Voltage Management for Piezoelectric Energy Harvesting

**DOI:** 10.3390/mi14081639

**Published:** 2023-08-20

**Authors:** Liao Wu, Xinhui Wang, Minghua Xie

**Affiliations:** School of Electronic Information and Electrical Engineering, Changsha University, Changsha 410022, Chinaxieminghua@126.com (M.X.)

**Keywords:** piezoelectric, energy harvesting circuit, DSSH, piezoelectric MEMS/NEMS

## Abstract

This paper presents a piezoelectric (PE) energy harvesting circuit based on the DSSH (double synchronized switch harvesting) principle. The circuit consisted of a rectifier and a DC–DC circuit, which achieves double synchronized switch operation for the PE transducer in each vibration half-cycle. One of the main challenges of the DSSH scheme was precisely controlling the switch timing in the second loop of the resonant loops. The proposed circuit included a MOS transistor in the second loop to address this challenge. It utilized its threshold voltage to manage the stored energy in the intermediate capacitor per vibration half-cycle to simplify the controller for the DSSH circuit. The circuit can operate under either the DSSH scheme or the ESSH (enhanced synchronized switch harvesting) scheme, depending on the value of the intermediate capacitor. In the DSSH scheme, the following DC–DC circuit reused the rectifier’s two diodes for a short period. The prototype circuit was implemented using 16 discrete components. The proposed circuit can be self-powered and started up without a battery. The experimental results showed that the proposed circuit increased the power harvested from the PE transducer compared to the full-bridge (FB) rectifier. With two different intermediate capacitors of 100 nF and 320 nF, the proposed circuit achieved power increases of 3.2 and 2.7 times, respectively. The charging efficiency of the proposed circuit was improved by a factor of 5.1 compared to the typical DSSH circuit.

## 1. Introduction

Wireless sensor nodes are widely used in various fields, such as medical implants, wildlife tracking, and pipeline monitoring [1]. However, frequent battery replacement or recharging poses a challenge for these applications. Vibration energy harvesting offers a promising solution for powering these wireless sensors. Among the different energy harvesting options, piezoelectric (PE) energy harvesting has garnered significant attention due to its high power density and scalability for PE transducers. To achieve an efficient power extraction and transfer to the load, PE energy harvesting circuits commonly employ rectification and maximum power point tracking. Designing rectifiers for power extraction and DC–DC circuits for power transfer presents unique challenges, which have been the main focus of the existing research efforts.

Extensive research has been conducted on PE energy harvesting circuits [2,3,4,5,6,7,8,9,10,11,12,13,14,15,16,17,18,19,20,21,22,23,24,25,26,27,28,29,30,31]. The primary objective of a PE energy harvesting circuit is to maximize the net power delivered to the load. It involves optimizing the power extraction from the PE transducer through the rectifier and the power transfer to the load via the DC–DC circuit. Efficiency is crucial, and minimizing circuit dissipation is a key factor. Figure 1 illustrates the block diagram of a PE energy harvesting circuit. Here, the PE transducer was modeled as a current source $i_{P}$ parallel to a capacitor $C_{P}$ and a resistor $R_{P}$, considering the weak coupling of small-scale PE transducers. The equation for $i_{P}$ is given by $i_{P}=I_{P}sin\left(2\pi f_{P}t\right)$, where $f_{P}$ is the excitation frequency of the PE transducer. This model served as the basis for the circuit analysis in this paper. The magnitude of the open-circuit voltage at the output of the PE transducer can be expressed as $V_{oc}=\frac{I_{P}}{2\pi f_{P}C_{P}}$. The rectifier converted the AC voltage output from the PE transducer into DC voltage, while a DC–DC circuit transferred the energy stored in the capacitor $C_{T}$ to the loads $C_{L}$ and $R_{L}$. Due to the relatively high impedance of PE transducers, including a significant capacitive term at the excitation vibration frequency, using a large inductor to match the capacitive term as suggested by the conjugate matching theorem was impractical. As a result, researchers have explored alternative schemes that employ a rectifier followed by a DC–DC converter, enabling an efficient power extraction and transfer. This approach addresses the challenges posed by PE transducer characteristics and contributes to advancing energy harvesting techniques.

Intensive research has been focused on the efficient rectification of PE AC voltages. The commonly used rectifier circuit is a full-bridge (FB) rectifier. To address diode losses, a voltage doubler (VD) rectifier with fewer diodes can be used. However, this type of rectifier has a drawback as it requires the internal capacitor $C_{P}$ of the PE transducer to be charged and discharged in each half-cycle, leading to wasted output charge. To reduce charge losses, the synchronized switching (SS) method, as described in [2,3], momentarily shorts the PE transducer at $t_{0}$ to discard the charge, resetting the capacitor voltage $V_{BA}$ to zero. The SS scheme does not require an inductor, thereby saving space. However, its efficiency is lower due to the discarded capacitor charge. In contrast, the synchronized switch harvesting on inductor (SSHI), such as the P-SSHI [4], S-SSHI [5], Hybrid SSHI [6], or Triple bias-flip SSHI [7], adopts an LC resonator with an external inductor. The LC resonator in the SSHI scheme changes the polarity of the capacitor charge at $t_{0}$, flipping the capacitor voltage $V_{BA}$ from positive to negative [8,9,10,11,12,13,14]. This allows the current $i_{P}$ to continuously charge the capacitor instead of discharging the negative charge before recharging. A common feature of the above rectifiers is that their output filtering capacitors are relatively large to maintain a stable rectified voltage. However, this also results in the output power of these rectifiers being load-dependent. To achieve maximum power, a second-stage DC–DC circuit is usually required to optimize the impedance for these rectifiers [15,16,17,18,19]. The DC–DC circuits often employ maximum power point tracking (MPPT) to achieve dynamic impedance matching. MPPT schemes necessitate extra control circuits for the DC–DC converter’s operation. The control circuits often dissipate a significant portion of the harvested energy, sometimes even exceeding the energy harvested. 

Researchers have proposed several alternative methods to address the issue of rectifier output power dependence on the load, including SECE [20], DSSH [21], and ESSH [22]. SECE is a representative scheme in which the LC resonator temporarily transfers the capacitor charge to the inductor at $t_{0}$ and then to the load [23,24,25,26,27,28,29]. The efficiency of the SECE scheme is independent of the load. The SECE circuit achieves four times the peak output power of the ideal FB rectifier. In contrast, the DSSH scheme transfers the capacitor charge to the intermediate capacitor at $t_{0}$ and then to the load. The DSSH scheme combines the advantages of the SSHI and SECE interface circuits. Under the same conditions, it achieves a higher output power than the SECE scheme while maintaining load-independent efficiency. The ESSH scheme is a further improvement based on the DSSH scheme. Wang et al. used a transformer rather than an inductor to implement a variation of the DSSH scheme [30]. Zou et al. used an inductor as the intermediate energy storage unit instead of a capacitor by inserting an additional resonant loop into the DSSH circuit [31]. These schemes commonly adopt two parts: a rectifier followed by a DC–DC circuit structure. Due to the circuit’s operation in a short period per vibration half-cycle, precise control timing for both parts is necessary. As a result, the controller is often complex, leading to a higher power dissipation.

This paper presents a self-powered DSSH circuit, incorporating a simplified low-power controller based on threshold voltage management. The main advantages of the circuit lie in its uncomplicated controller and reduced drive losses for the MOSFET in the DC–DC circuit. The proposed circuit utilizes the MOSFET’s threshold voltage to automatically switch resonant loops in the DSSH circuit. The operational mode, whether DSSH or ESSH, depends on the value of the intermediate energy storage capacitor. The circuit prototype consists of 16 discrete components, eliminating the need for a complex controller. Additionally, the circuit can cold-start even when the battery is drained. 

This paper is organized as follows. Section 2 describes the basic operation of the DSSH circuit and discusses the optimization of controller power dissipation. Section 3 presents the proposed DSSH circuit, including its operation, modeling, analysis, and the identification of the value range for the intermediate capacitor $C_{T}$. Section 4 covers the implementation of the proposed circuit and describes how it operates. Section 5 shows the measurement results of the proposed circuit. Section 6 concludes the paper.

## 2. Preliminaries

### 2.1. Review of the DSSH Circuit and Relevant Works

Figure 2 shows a typical DSSH circuit [21], which is divided into two parts. The first part adopts an S-SSHI circuit topology with an intermediate energy storage capacitor $C_{T}$. The second part includes a switch ${SW}_{2}$, an inductor $L_{2}$, a diode $D_{1}$, and a load $R_{L}$ and $C_{L}$, forming a buck-boost topology.

As shown in Figure 2, the DSSH circuit operates briefly per vibration half-cycle. The circuit sequentially activates three resonant loops, transferring the energy accumulated in the internal capacitor $C_{P}$ of the PE transducer to the load. One of the design challenges of the DSSH circuit is controlling the switching timing of the switch ${SW}_{2}$. It needs to be turned on after the switch ${SW}_{1}$ turns off and turned off promptly after completing the energy extraction from the capacitor $C_{T}$, i.e., when the voltage in the capacitor $C_{T}$ decreases to zero. In a previous study [21], the DSSH circuit was implemented using DSPACE, but the circuit’s self-powered functionality was not achieved. To address the cold-start issue, Shen et al. introduced an additional start-up circuit with assistance from PZT. Furthermore, the complexity of the controller stemmed from its management of ${SW}_{2}$, which involved tasks such as control signal generation, threshold control, and power management. These controller modules resulted in a higher power dissipation. Therefore, their proposed system operation was divided into two states: charging and operating [22]. To mitigate this power demand, Wang et al. and Zou et al. adopted an intermittent power supply approach [30,31]. However, their circuits relied on variable resistors to adjust the timing circuit. Note that all the mentioned circuits utilized comparator outputs to drive the MOSFET.

This paper adopted the method presented in [30,31] for the DSSH circuit, where the control circuit was activated only when necessary. However, the proposed method utilized the charge accumulated during the charging of the intermediate energy storage capacitor $C_{T}$ to drive the MOSFET in the DC–DC circuit of the DSSH circuit. This greatly simplified the controller. Furthermore, depending on the capacitor’s value, the circuit could operate in either the DSSH or ESSH scheme. The feasibility of adapting these two schemes was attributed to the shared circuit topology between the DSSH and ESSH circuits.

### 2.2. Optimizing the Losses of the DSSH Circuit Controller

Since the primary function of the DSSH circuit controller is to control the circuit switching on and off, the power dissipation can be divided into two parts: the losses in the control circuit itself and the losses associated with driving the MOSFET gate. The power losses of the DSSH circuit controller can be expressed as the following.
(1)$$P_{ctrl}=P_{gate}+P_{D}=2V_{DD}Q_{G}f_{P}+I_{D}V_{DD}$$
where $V_{DD}$ represents the supply voltage of the control circuit, $Q_{G}$ is the total gate charge for driving the MOSFET, and $I_{D}$ represents the total current dissipated by the control circuit when the gate driver is inactive. As aforementioned, in order to reduce $P_{ctrl}$, it is essential to minimize both the losses in the control circuit and the MOSFET’s drive losses. This necessitates simplifying the controller, such as the peak voltage detection circuits, timing circuits, and the DC–DC control circuit. Additionally, efforts should be made to decrease the switch drive losses.

Unlike the typical DSSH circuit, the proposed circuit incorporated a MOSFET into the second loop of the double synchronized resonant loops. This MOSFET utilized its threshold voltage to manage the stored energy in the intermediate capacitor during each vibration half-cycle. The method adopted in this paper had two advantages. (1) It could utilize the MOSFET’s inherent voltage threshold to automatically switch the resonant loops of the DSSH circuit, thereby reducing the control losses. (2) It enabled the extraction of the energy that drives the MOSFET, thus avoiding losses to drive the gate of the MOSFET. Moreover, a highly simplified control circuit operated only when necessary to reduce the control losses $P_{D}$. 

## 3. Proposed Energy Harvesting Circuit

### 3.1. Block Diagram

Figure 3 illustrates the block diagram of the proposed circuit, which included two parts. The first part followed the S-SSHI scheme and served as rectification, extracting energy from the PE transducer. It comprised two switches $S_{1}$ and $S_{2}$, two diodes $D_{1}$ and $D_{2}$, and an inductor $L_{1}$ connected in series with the PE transducer. The second part employed a boost topology account for transferring the extracted energy to the load. Notably, $C_{T}$ acted as a temporary intermediate capacitor. For simplicity in the analysis, the diodes and switches were assumed to be ideal unless stated otherwise.

### 3.2. Operation

Figure 4 illustrates the basic operation of the proposed circuit. Figure 4a shows the waveforms of the transducer current $i_{P}$, PE transducer voltage $V_{BA}$, inductor currents $i_{L1}$ and $i_{L2}$, and capacitor voltage $V_{T}$. Figure 4b–e depict the states of the switches for specific time intervals. During the interval $t_{0}$ < t < $t_{1}$, as shown in Figure 4b, the transducer current $i_{P}$ was positive, and the switches $S_{1}$, $S_{2}$, and $S_{3}$ were open. The inductor current $i_{L}$ was zero. During this period, the transducer current charged the internal capacitor $C_{P}$, increasing $V_{BA}$. At $t_{1}$, $V_{BA}$ (=$V_{B}$) reached its peak, causing $S_{1}$ to close and create a resonant loop along the blue dot line, as shown in Figure 4c, to perform the SSHI operation. The energy stored in $C_{P}$ was transferred to the inductor, and then the inductor energy was transferred back to the capacitor. As the current passed through the capacitor $C_{T}$, $C_{T}$ stored the energy during this period, causing the voltage of $C_{T}$ to increase. When the resonant loop’s current decreased to zero at $t_{2}$, the switch $S_{1}$ opened, and $S_{3}$ closed. The subsequent DC–DC circuit began operating to transfer the energy stored in $C_{T}$ to the load. A resonant loop was formed through the devices $C_{T}$-$S_{3}$-$L_{2}$-$M_{1}$-$C_{T}$ during $t_{2}$ < t < $t_{3}$. As the MOSFET $M_{1}$ was inserted into the loop, when the capacitor voltage $V_{T}$ decreased below the threshold voltage of MOSFET $M_{1}$, $M_{1}$ opened at $t_{3}$. At $t_{3}$, the inductor current $i_{L2}$ reached its peak $I_{M}$. Then, the circuit established a freewheeling path through the devices $C_{T}$-$S_{3}$-$L_{2}$-$D_{3}$-$R_{L}$/$C_{L}$ for the inductor current to charge the load. Two scenarios were considered based on whether the energy in capacitor $C_{T}$ was depleted. (1) If $C_{T}$ contained sufficient energy after the inductor current $i_{L2}$ drops to zero, there was still residual energy in $C_{T}$. In this case, the circuit operated in the ESSH mode. (2) If the energy stored in $C_{T}$ was insufficient before the inductor current drops to zero, the energy in $C_{T}$ was depleted and the voltage $V_{T}$ reached zero at $t_{4}$, establishing another freewheeling path through the devices $D_{1}$-$D_{2}$-$S_{3}$-$L_{2}$-$D_{3}$-$R_{L}$/$C_{L}$ for the inductor current to charge the load. In this case, the circuit operated in the DSSH mode. The same process was repeated for the negative transducer current. It is worth noting that due to the use of the intermediate capacitor $C_{T}$ to drive MOSFET $M_{1}$, the circuit incurred no drive losses $P_{gate}$, as shown in Equation (1).

### 3.3. Modeling and Analysis

The proposed circuit ensured an optimal harvested power regardless of the load connected to the PE transducer. Now, let’s analyze the resonant loops using ideal components. Assuming the polarity of the PE transducer current $i_{P}$ changed, either switch $S_{1}$ or $S_{2}$ was conducted immediately. Additionally, switch $S_{3}$ was closed when the peak voltage of the capacitor $C_{T}$ was detected. During the analysis, $V_{D}$ modeled the total forward voltage decrease across all the components in the resonant loops, while R represented the total parasitic resistances along those resonant loops. The PE transducer model’s $R_{P}$ was neglected due to its large value. 

First resonant loop:

Figure 5 illustrates the first resonant loop for extracting energy from the PE transducer (Figure 4c). In this loop, the switches SW and L corresponded to $S_{1}$ or the $S_{2}$ and $L_{1}$, respectively, as shown in Figure 3. When SW was closed, applying Kirchhoff’s voltage law resulted in the following differential equation.
(2)$$LC_{S}\frac{d^{2}i_{L}}{dt^{2}}+RC_{S}\frac{di_{L}}{dt}+i_{L}=0$$
where $C_{S}=\frac{C_{T}C_{P}}{C_{T}+C_{P}}$. At t = 0, $v_{P}\left(0\right)=V_{M}$, $v_{T}\left(0\right)=V_{E}$, and $i_{L}\left(0\right)=0$. The inductor current $i_{L}$, the capacitor voltage $v_{P}$, i.e., $V_{BA}$, and the capacitor voltage $v_{T}$, i.e., $V_{T}$ were readily obtained as the following.
(3)$$\;i_{L}(t)=\frac{V_{M}-V_{E}-V_{D}}{\omega L}e^{-\beta t}sin\omega t$$
(4)$$\;v_{P}(t)=\frac{C_{P}V_{M}+C_{T}V_{E}}{C_{P}+C_{T}}+\frac{C_{T}V_{D}}{C_{P}+C_{T}}+\frac{C_{T}\left(V_{M}-V_{E}-V_{D}\right)}{C_{P}+C_{T}}\cdot {\frac{\omega_{o}}{\omega }e}^{-\beta t}cos\left(\omega t-\phi \right)$$
(5)$$v_{T}(t)=\frac{C_{P}V_{M}+C_{T}V_{E}}{C_{P}+C_{T}}-\frac{C_{P}V_{D}}{C_{P}+C_{T}}-\frac{C_{P}\left(V_{M}-V_{E}-V_{D}\right)}{C_{P}+C_{T}}\cdot {\frac{\omega_{o}}{\omega }e}^{-\beta t}cos\left(\omega t-\phi \right)$$
where $\beta =\frac{R}{2L}$, $\omega =\sqrt{\omega_{o}^{2}-\beta^{2}}$, $\omega_{o}=\frac{1}{\sqrt{LC_{S}}}$, and $\phi ={tan}^{-1}\left(\beta /\omega \right)$.

As shown in Figure 5b, during the time interval from $t_{1}$ to $t_{2}$, which lasted for a duration of $t_{a}$($=\frac{\pi }{\omega }$), the inductor current returned to zero. At $t_{2}$, the switch SW opened and the voltage $v_{P}$ and $v_{T}$ were obtained using the following.
(6)$$\;V_{m}=v_{P}\left(\frac{\pi }{\omega }\right)=V_{M}-\frac{C_{T}\left(V_{M}-V_{E}-V_{D}\right)}{C_{P}+C_{T}}\left(1+\lambda \right)$$
(7)$$\;V_{T,PK}=v_{T}\left(\frac{\pi }{\omega }\right)=V_{E}+\frac{C_{P}\left(V_{M}-V_{E}-V_{D}\right)}{C_{P}+C_{T}}\left(1+\lambda \right)$$
where $\lambda =e^{-\frac{\pi }{2\sqrt{Q^{2}-\frac{1}{4}}}}$ and $Q=\frac{1}{R}\sqrt{\frac{L}{C}}$. From Equations (6) and (7), during the time interval $t_{1}$ < *t* < $t_{2}$, energy was transferred from $C_{P}$ to $C_{T}$, causing a decrease in the voltage $v_{P}$ and increase in the voltage $v_{T}$. If $C_{T}$ is much larger than $C_{P}$, $v_{P}$ would decrease by a larger magnitude, while $v_{T}$ would increase by a smaller magnitude. Conversely, if $C_{T}$ is much smaller than $C_{P}$, the PE voltage may not achieve a voltage flipping. Therefore, an appropriate range existed for the value of $C_{T}$ to enable voltage flipping. In this study, we set the value of the capacitor $C_{T}$ in the steady state to achieve PE voltage flipping. To ensure voltage flipping, we needed to satisfy $V_{m}>0$ per half vibration cycle, as indicated by the equation $2V_{oc}-V_{m}=V_{M}$. We obtained the following relationship that the value of $C_{T}$ needed to satisfy.
(8)$$C_{T}>\frac{C_{P}}{\lambda -\frac{\left(V_{E}+V_{D}\right)}{2V_{oc}}\left(1+\lambda \right)}$$
where $V_{D}$ is the first resonant loop.

Second resonant loop:

Figure 6 illustrates the second resonant loop used for the energy transfer (Figure 4d). In this resonant loop, a MOSFET $M_{1}$ was inserted to control the charge in the capacitor $C_{T}$. In the following analysis, $M_{1}$ was modeled as a simple threshold-based switch, which turned on when $V_{GS}$ > $V_{GS(TH)}$ and turned off vice versa. Here, the switch SW and inductor L corresponded to $S_{3}$ and $L_{2}$, respectively, as shown in Figure 3. 

When $V_{T,PK}>V_{GS(TH)}$ and SW was closed, we applied Kirchhoff’s voltage law to derive the following differential equation for the circuit shown in Figure 6.
(9)$$LC_{T}\frac{d^{2}i_{L}}{dt^{2}}+RC_{T}\frac{di_{L}}{dt}+i_{L}=0$$

At $t_{2}$, the capacitor voltage of $C_{T}$ was $V_{T,PK}$ and the inductor current of L was zero. The capacitor voltage $v_{T}$ and the inductor current $i_{L}$ were obtained as the following.
(10)$$v_{T}\left(t\right)=\left(V_{T,PK}-V_{D}\right)\frac{\omega_{o}}{\omega }e^{-\beta t}cos\left(\omega t-\phi \right)+V_{D}$$
(11)$$\;i_{L}\left(t\right)=\frac{V_{T,PK}-V_{D}}{\omega L}e^{-\beta t}sin\omega t$$

Since the energy stored in the capacitor $C_{T}$ was transferred to the inductor L, the inductor current $i_{L}$ increased and capacitor voltage $v_{T}$ decreased. When $v_{T}$ decreased to $V_{GS(TH)}$ and $i_{L}$ reached its peak value of $I_{M}$ at t = $t_{3}$, MOSFET $M_{1}$ turned off, interrupting the resonant loop. At $t_{3}$, the remaining energy in $C_{T}$ could be written as the following.
(12)$$E_{T}=\frac{1}{2}C_{T}v_{T}{\left(t_{3}\right)}^{2}=\frac{1}{2}C_{T}V_{GS\left(TH\right)}^{2}$$

The energy transferred to the inductor L could be written as the following.
(13)$$E_{L2}=\frac{1}{2}Li_{L}{\left(t_{3}\right)}^{2}=\left(\frac{1}{2}C_{T}V_{T,PK}^{2}-\frac{1}{2}C_{T}V_{GS\left(TH\right)}^{2}\right)\cdot \eta$$
where η represents the efficiency of the energy transfer from the capacitor $C_{T}$ to the inductor L.

Third resonant loop:

Figure 7 shows the third resonant loop used for the power transfer. In this loop, L represented the $L_{2}$ and $V_{L}$ denoted the load voltage, as shown in Figure 3. By solving the differential equations with the initial voltage and current, the capacitor voltage $v_{T}$ and the inductor current $i_{L}$ were obtained as follows.
(14)$$v_{T}\left(t\right)=V_{D}+V_{L}+A\omega_{o}Le^{-\delta t}sin\left(\omega t+\varphi +\psi \right)$$
(15)$$i_{L}\left(t\right)=Ae^{-\delta t}sin\left(\omega t+\varphi \right)$$
where $A=\sqrt{{I_{M}}^{2}+{\left(\frac{\left(V_{GS(TH)}-V_{D}-V_{L}\right)-\frac{R}{2}{\cdot I}_{M}}{\omega L}\right)}^{2}}$, $\varphi =arcsin\left(\frac{I_{M}}{\sqrt{{I_{M}}^{2}+{\left(\frac{\left(V_{GS(TH)}-V_{D}-V_{L}\right)-\frac{R}{2}\cdot I_{M}}{\omega L}\right)}^{2}}}\right)$, and $\omega =\sqrt{\omega_{o}^{2}-\delta^{2}}$, $\delta =\frac{R}{2L}$, $\omega_{o}=\frac{1}{\sqrt{LC_{T}}}$.

When $i_{L}\left(t_{5}\right)=0$, $v_{T}\left(t_{5}\right)>0$ in a steady state, as shown in Figure 4a, residual energy was still stored in $C_{T}$ after charging the load. In this case, the circuit operated in the ESSH scheme.

On the other hand, when $v_{T}\left(t_{4}\right)=0$, $i_{L}\left(t_{4}\right)>0$, as shown in Figure 4a, the energy stored in the capacitor $C_{T}$ was fully discharged to the load, but there some energy still remained in the inductor. The inductor, as shown in Figure 4e, established an alternative freewheeling path through the devices $D_{2}$-$D_{1}$-$S_{3}$-$L_{2}$-$D_{3}$-$R_{L}$/$C_{L}$, allowing for the inductor current to charge the load. 

Fourth resonant loop:

As shown in Figure 4e, the fourth resonant loop utilized the two diodes $D_{1}$ and $D_{2}$ of the rectifier to discharge the inductor energy. This freewheeling path resembled the branch of a buck-boost circuit. Since all the energy stored in $C_{T}$ per half vibration cycle was transferred to the load, the circuit operated in the DSSH scheme.

### 3.4. Analysis of the Value Range for Capacitor $C_{T}$

Let’s define $C_{T}=xC_{P}$. To operate the circuit in the DSSH scheme, it needed to satisfy Equation (8) and $E_{L2}$ (Equation (13)) > $E_{T}$ (Equation (12)). In contrast, for the ESSH scheme, it needed to satisfy $V_{T,PK}>V_{GS\left(TH\right)}$ and $E_{L2}$ < $E_{T}$. The permissible range of values for x were calculated, as shown in Table 1. Here, $V_{D}$ represented the total voltage decrease across the first resonant loop.

## 4. Circuit Implementation

Figure 8 shows the implementation of the proposed circuit, consisting of two main parts: the rectifier and the DC–DC circuit. In the rectifier, a self-powered switch was employed, comprising two PNP transistors ($Q_{2}$ and $Q_{4}$), two NPN transistors ($Q_{1}$ and $Q_{3}$), and a capacitor ($C_{1}$). Its main function was to detect the peak voltage of the PE transducer output and control the switch operation of either $Q_{3}$ or $Q_{4}$. On the other hand, the DC–DC circuit incorporated a self-powered peak detection circuit using the components $D_{3}$, $Q_{5}$, and $Q_{6}$. This circuit’s purpose was to detect the peak voltage of the capacitor $C_{T}$, thereby facilitating the operation of the switch in the DC–DC circuit. The working principle is briefly described below.

When the transducer current $i_{P}$ was positive, it charged the capacitor $C_{P}$, and the NPN transistor $Q_{1}$’s base emitter was turned on to connect the capacitors $C_{1}$ and $C_{P}$ in parallel. Since $C_{1}$ was much smaller than $C_{P}$, it consumed only a small portion of energy from the PE transducer. When the current $i_{P}$ crossed the zero point and began to discharge $C_{P}$, the voltage $V_{BA}$ decreased. However, the voltage across the capacitor $C_{1}$ remained unchanged. It caused $Q_{1}$ to turn off and $Q_{2}$ to turn on. As a result, $Q_{3}$ was turned on, forming the first resonant loop through the components $C_{P}$-$Q_{3}$-$D_{3}$-$C_{T}$-$D_{2}$-$C_{P}$. The energy stored in $C_{P}$ was transferred through inductor $L_{1}$ to the capacitor $C_{T}$ until the inductor current $L_{1}$ returned to zero. During this process, the current flowing through diode $D_{3}$ caused a forward voltage decrease, which forced $Q_{5}$ and $Q_{6}$ to turn off. Once the current in the first resonant loop decreased to zero, and $C_{T}$’s voltage exceeded $V_{GS\left(TH\right)}$, a MOSFET $M_{1}$ turned on. With no current flowing through $D_{3}$, $Q_{5}$ and $Q_{6}$ turned on, forming the second resonant loop through the components $C_{T}$-$Q_{5}$-$Q_{6}$-$L_{2}$-$M_{1}$-$C_{T}$. The inductor current $i_{L2}$ began to increase from zero, and the voltage across the capacitor $C_{T}$ began to decrease. The energy stored in $C_{T}$ was transferred to inductor $L_{2}$. When the capacitor voltage $V_{T}$ decreased to $V_{GS\left(TH\right)}$, $M_{1}$ turned off. The remaining energy in the inductor was discharged through a resonant loop via the components $C_{T}$-$Q_{5}$-$Q_{6}$-$L_{2}$-$D_{4}$-$R_{L}$/$C_{L}$-$C_{T}$ (and $D_{2}$-$D_{1}$-$Q_{6}$-$L_{2}$-$D_{4}$-$R_{L}$/$C_{L}$-$D_{2}$) until the inductor current returned to zero. 

To verify the flexibility of the proposed circuit to operate in either the DSSH or ESSH mode, we conducted simulations using two different intermediate capacitors, 100 nF and 320 nF. The simulations were conducted under the same excitation conditions, using the PE transducer model shown in Figure 1 and the simulation parameters listed in Table 2. The obtained waveforms are presented in Figure 9. Figure 9a shows that the voltage $V_{T}$ increased from 0.4 V, which was not 0 V, due to the effect of the forward voltage decrease in the resonant loop. This observation indicated that the circuit operated in the DSSH scheme when the intermediate capacitor was set to 100 nF. Conversely, as shown in Figure 9b, the voltage $V_{T}$ increased to 2.875 V from 1.4 V and then decreased to 1.4 V. This behavior confirmed that the circuit operated in the ESSH scheme when the intermediate capacitor was set to 320 nF. The reason for the smaller peak current of IL1 in the circuit with a 100 nF intermediate capacitor compared to the $i_{L1}$ peak current of the 320 nF intermediate capacitor was due to the smaller difference between $V_{M}$ and $V_{E}$, as indicated by Equation (3).

During the circuit simulation, we examined the power dissipation breakdown. The simulation used components that closely matched those used in the experiments described in Section 5. Table 3 shows the component conduction losses of the circuit using 100 nF and 320 nF intermediate capacitors under the same excitation. Whether the circuit operated under the DSSH or ESSH schemes, the highest losses were from $Q_{3}$ and $Q_{4}$. This losses primarily originated from the conduction loss caused by the forward voltage decrease $V_{CE}$ between the collectors and emitters in $Q_{3}$ and $Q_{4}$ when they are operating in the saturation region. Therefore, there is still room for improvement in the first part of the proposed circuit. However, this paper primarily focused on reducing the power dissipation of the second part of the DSSH circuit. 

## 5. Experimental Results

The proposed circuit was implemented on a PCB board using 16 discreate components. Off-the-shelf inductors were employed in the experiment. The inductor $L_{1}$ (model number ASPI-0403S-152M) had a value of 1.5 mH and a DC resistance (RDC) of 4.2 Ω. The inductor $L_{2}$ (model number CDRH10D68R) had a value of 1 mH and a RDC of 1.56 Ω. The three PNP transistors, $Q_{2}$, $Q_{4}$, and $Q_{5}$, were 2N3906s, with a *V_BE(SAT)_* voltage of −0.95 V and a *V_CE(SAT)_* voltage of −0.4 V. The three NPN transistors, $Q_{1}$, $Q_{3}$, and $Q_{6}$, were 2N3904s, with a *V_BE(SAT)_* voltage of 0.95 V and a *V_CE(SAT)_* voltage of 0.3 V. The MOSFET had a threshold voltage ranging from 1.2 V to 2.5 V, and a *V_DS_* voltage of 40 V. The capacitance $C_{L}$ was 1 μF, and the load resistor $R_{L}$ was 100 kΩ, unless otherwise specified in this paper.

A piezoelectric cantilever (MIDE, PPA-1021) with a tip mass of 1.6 g was used for the measurements. The cantilever was placed on a thick aluminum plate, as shown in Figure 10. An accelerometer module with ADXL345 attached to the shaker was used to measure the acceleration. The internal capacitance of the PE cantilever was measured to be 22 nF, and the resonant frequency for the PE cantilever with the attached mass was measured to be 65 Hz.

The first experiment was performed to test the proposed circuit’s functionality. The shaker oscillated at a frequency of 65 Hz with an RMS acceleration of 0.038 g. The open-circuit voltage amplitude was measured as 3.28 V using an oscilloscope. Initially, we applied a load of 530 nF and 100 kΩ to the rectifier only to test its operation. Figure 11a shows the output voltage of the PE cantilever, indicating the successful operation of the first resonant loop as the PE voltage decreased from 2 V to −1.6 V or increased from −2 V to 1.58 V. It was essential to note that the activation time of the first resonant loop was not precisely synchronized with the peak output voltage of the PE cantilever due to the non-ideality of the discrete components. Next, we removed the load from the rectifier and connected it to the DC–DC circuit. For both parts, we selected a 100 nF intermediate capacitor and a load consisting of a of a 1 μF capacitor $C_{L}$ and a 100 kΩ resistor $R_{L}$. The PE voltage is shown in Figure 11b. The PE voltage changed from positive to negative, decreasing from 2 V to −0.6 V, and from negative to positive, increasing from −2 V to +0.6 V. The reason for the change in the PE cantilever voltage was the replacement of the rectifier output capacitor from a load capacitor of 500 nF to an intermediate capacitor of 100 nF in the DSSH circuit.

The second experiment was conducted to examine the impact of different intermediate capacitors on the PE transducer’s output voltage in a steady state. Figure 12 illustrates the results of the experiment. When the intermediate capacitor was set to 22 nF, as analyzed in Section 3.3, the output voltage of the PE transducer did not exhibit significant voltage flipping, compared to the case with 100 nF, as shown in Figure 11b. This was because the value of the intermediate capacitor was comparable to the internal capacitor of the PE transducer. However, large voltage flipping occurred when the intermediate capacitor was increased to 320 nF, as shown in Figure 12b. The PE voltage shown in Figure 12b changed from positive to negative, decreasing from 4.4 V to −2.8 V, and from negative to positive, increasing from −4.4 V to 2.8 V. The different PE cantilever voltage flipping was due to the different values of $C_{T}$ and the different residual charges stored in the capacitor $C_{T}$ during each vibration half-cycle.

The third experiment tested the cold start of the proposed circuit. We used an intermediate capacitor of 100 nF and a load resistor $R_{L}$ of 100 kΩ for the test. We controlled the shaker’s vibration acceleration by adjusting the signal generator’s amplitude. When the measured open-circuit voltage of the PE transducer was 2.2 V, we connected the PE transducer’s output to the proposed circuit. The circuit’s output voltage $V_{L}$ is shown in Figure 13a. In the figure, it can be observed that when the shaker was turned on, the voltage $V_{L}$ increased from 0 V to approx. 0.23 V and then decreased to approximately 0.1 V, indicating that the circuit failed to start. However, after increasing the vibration magnitude, the circuit successfully started, as shown in Figure 13b–d. The circuit’s output voltage $V_{L}$ gradually increased from 0 V to 0.92 V over 0.6 s, as shown in Figure 13b. The sawtooth waveform of $V_{L}$ indicated that the energy was transferred to the load in a very short period when the PE voltage reached its peak during each vibration half-cycle. The measured output voltage of the PE transducer is shown in Figure 13c, and a zoomed-in view of the waveform is provided in Figure 13d. As shown in Figure 13d, after three cycles, the resonant loop began to function, resulting in the flipping of the PE voltage.

Figure 14 shows the measured output power of the proposed circuit at various load resistors under the same excitation condition, using two different intermediate capacitors with values of 100 nF and 320 nF for testing. It can be observed that the output power decreased as the load resistor $R_{L}$ increased. This decrease was attributed to a higher load voltage $V_{L}$, leading to higher power losses in the proposed circuit. Moreover, the output power of the proposed circuit with a 320 nF intermediate capacitor was lower than that with a 100 nF capacitor. This was because, as shown in Figure 4a, the switch ${SW}_{2}$ in the second resonant loop of the proposed circuit only turned on at time $t_{2}$ when the inductor current $i_{L1}$ was zero, representing a specific case of the ESSH scheme in operation for the proposed ESSH circuit. The RMS current in the first resonant loop of the DSSH circuit with a 320 nF intermediate capacitor was higher than that with a 100 nF intermediate capacitor, as indicated in Figure 9. We also compared our circuit with the FB rectifier, achieving a 3.2 times higher output power using a 100 nF intermediate capacitor and a 2.7 times higher output power using a 320 nF intermediate capacitor.

Figure 15 shows the time-dependent output voltage of the proposed circuit using a load of a 50 V 10 mF electrolytic capacitor. The shaker was oscillated at 65 Hz with an RMS acceleration of 0.062 g. The peak open-circuit voltage of the PE cantilever was observed as 5.4 V. The figure also shows the output voltage of another design, a conventional DSSH circuit. This typical DSSH circuit differed from the proposed circuit in that the second part of the DSSH circuit employed a buck-boost topology, and the second resonant loop excluded the MOSFET $M_{1}$. In both cases, the intermediate capacitor had a value of 100 nF. The proposed circuit’s output voltage attained 445 mV at 860 s, while the typical DSSH circuit reached 87.3 mV at 300 s but does not sustain a further increase. Hence, considering the final charging voltage, the proposed circuit outperformed the typical DSSH circuit by a factor of 5.1 or potentially more. This enhancement can be attributed to the boost topology implemented in the proposed DSSH circuit.

Table 4 summarizes the performance and characteristics of recent state-of-the-art DSSH circuits. The charging efficiency, defined as the ratio of the final charged voltage achieved by the proposed circuit to that of the typical DSSH circuit under identical excitation conditions, was a metric for comparison. However, making a direct and fair comparison to other circuits was difficult due to using different PE transducers, excitation conditions, and different circuit parameters. Among the five circuit designs shown in Table 4, all of them were built using discrete components, but the proposed circuit exhibited the highest charging efficiency. The circuits achieved a self-powered state except for [21]. However, the proposed circuit employed the fewest components, 16 in total. In terms of the overall design complexity, our circuit’s implementation was the simplest. It’s worth emphasizing that the primary power dissipation for the proposed circuit was associated with components $Q_{3}$ and $Q_{4}$, as shown in Figure 8. Implementing these components using integrated circuits (ICs) had the potential to significantly enhance the circuit’s overall performance. In summary, the proposed circuit’s key advantage was in utilizing the MOSFET threshold voltage for power management, which simplified the controller and reduced the power dissipation.

In the end, we provided a general design rule for optimizing DSSH circuits. As shown in Figure 3, a general DSSH circuit utilizes three resonant loops per half-cycle to transfer energy from the PE transducer to the load. In the following analysis, the variables $R_{1(2,3),total}$, $V_{F1(2,3),total}$, and $P_{cond,1(2,3)}$ represent the cumulative values of the resistances, the total forward voltage decreases, and the conduction losses across each resonant loop, respectively. Through our calculations, the conduction losses for each resonant loop within in the DSSH circuit were obtained as follows.
(16)$$P_{cond,1}=2f_{P}\left(\int_{t_{1}}^{t_{2}}i_{L1}\left(t\right)\cdot V_{F1,total}dt+\int_{t_{1}}^{t_{2}}i_{L1}{\left(t\right)}^{2}\cdot R_{1,total}dt\right)=\left[2Q_{1}V_{F1,total}+I_{L1,PK}^{2}R_{1,total}\left(t_{2}-t_{1}\right)\right]f_{P}$$
(17)$$P_{cond,2}=2f_{P}\int_{t_{2}}^{t_{3}}i_{L2}{\left(t\right)}^{2}\cdot R_{2,total}dt=I_{L2,PK}^{2}R_{2,total}\left(t_{3}-t_{2}\right)f_{P}$$
(18)$$P_{cond,3}=2f_{P}\left(\int_{t_{3}}^{t_{4}}i_{L2}\left(t\right)\cdot V_{F3,total}dt+\int_{t_{3}}^{t_{4}}i_{L2}{\left(t\right)}^{2}\cdot R_{1,total}dt\right)=\left(V_{VF3,total}+\frac{2}{3}I_{L2,PK}R_{3,total}\right)I_{L2,PK}\left(t_{4}-t_{3}\right)f_{P}$$
where $i_{L1}\left(t\right)$ and $i_{L2}\left(t\right)$ are the currents flowing through inductors $L_{1}$ and $L_{2}$, respectively. $Q_{1}$ is the accumulated charge from time $t_{1}$ to $t_{2}$. $i_{L1,PK}$ and $i_{L2,PK}$ are the current peaks of $i_{L1}\left(t\right)$ and $i_{L2}\left(t\right)$, respectively. 

In addition to conduction losses, the DC–DC circuit within the DSSH circuit generated switching losses. The switching losses included losses due to voltage and current overlap during the switch transition, along with losses caused by the capacitance at the switch node during its ON transition. Hence, the losses incurred from the switching transitions of the DC–DC circuit can be expressed as the following.
(19)$$P_{MOSFET,switch}=V_{DRAIN}I_{L1,PK}t_{f}f_{P}+C_{DRAIN}V_{DRAIN}^{2}f_{P}=\left(I_{L1,PK}t_{f}+C_{DRAIN}V_{DRAIN}\right)V_{DRAIN}f_{P}$$
where $C_{DRAIN}$ represents the parasitic capacitance at the switch node and $V_{DRAIN}$ is the voltage to which $C_{DRAIN}$ is charged when the MOSFET is turned off.

Therefore, the total losses in a DSSH circuit can be expressed as the following.
(20)$$P_{total}=\sum_{i=1}^{3}P_{cond,i}+P_{MOSFET,switch}$$

Using Equations (16) and (20), this paper provided a set of recommended general optimized design rules for a general DSSH circuit.

Minimize the forward voltage decreases generated by the resonant loops used in the DSSH circuit.Minimize the resistances within the resonant loops used in the DSSH circuit.Carefully design the switches and diodes in the following DC–DC circuit of the DSSH circuit to minimize *C_DRAIN_*.Reducing the loop current in the resonant loops is beneficial for further reducing losses.

Following these steps can result in an enhanced performance of DSSH circuits.

## 6. Conclusions

This paper presented a self-powered piezoelectric energy harvesting circuit based on a DSSH scheme. The proposed circuit incorporated a MOS transistor into the second loop of the double synchronized resonant loops. It provided the advantage of utilizing the MOSFET’s threshold voltage to manage the stored energy in the intermediate capacitor per half vibration cycle, simplifying the controller for the switching time. Depending on the value of the intermediate capacitor, the circuit could operate under either the DSSH scheme or the ESSH scheme, leveraging the same circuit topology for both schemes. The prototype circuit, implemented using 16 discrete components, demonstrated a self-powered functionality and the capability to start without the need for an external battery. The experimental results showed that the proposed circuit increased the power harvested from the PE transducer compared to the full-bridge rectifier. Using two different intermediate capacitors of 100 nF and 320 nF, the proposed circuit achieved power increases of 3.2 and 2.7 times, respectively. The charging efficiency of the proposed circuit was improved by a factor of 5.1 compared to the typical DSSH circuit. Future work will focus on implementing the proposed scheme using an integrated circuit, which can help reduce the power dissipation by utilizing active diodes with low-power controllers. Moreover, integrated circuits will help mitigate the adverse effects caused by non-idealities in discrete devices. 

## Figures and Tables

**Figure 1 micromachines-14-01639-f001:**
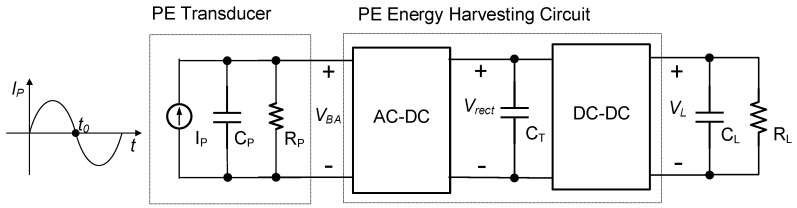
Block diagram for a PE energy harvesting circuit.

**Figure 2 micromachines-14-01639-f002:**
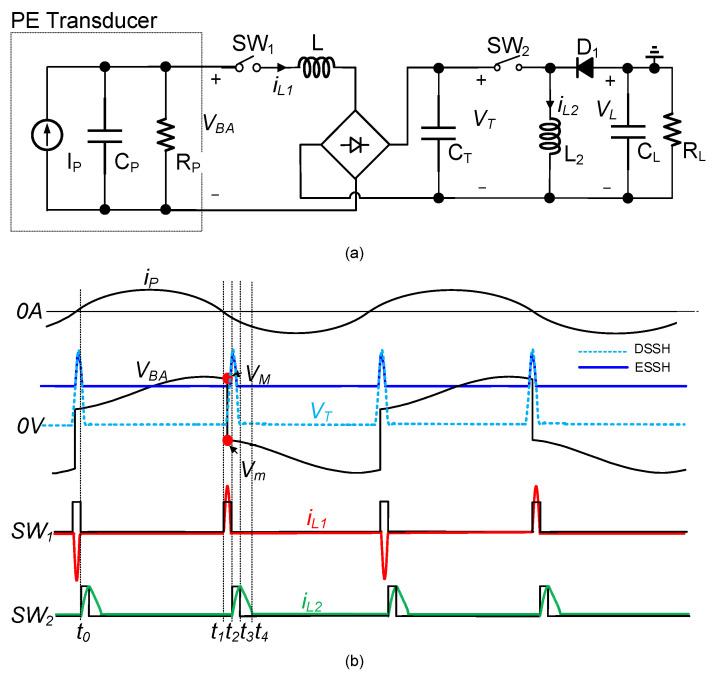
(**a**) Typical DSSH circuit; (**b**) simplified waveforms.

**Figure 3 micromachines-14-01639-f003:**
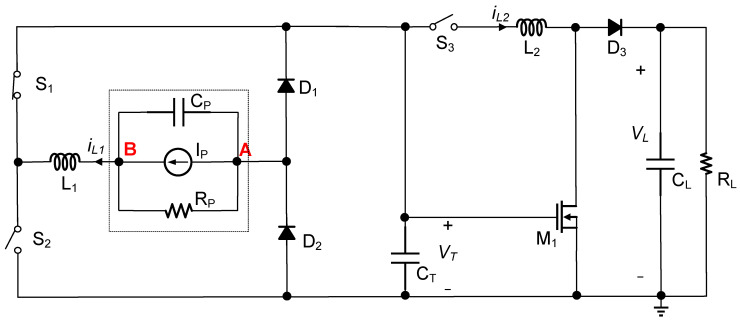
Proposed PE energy harvesting.

**Figure 4 micromachines-14-01639-f004:**
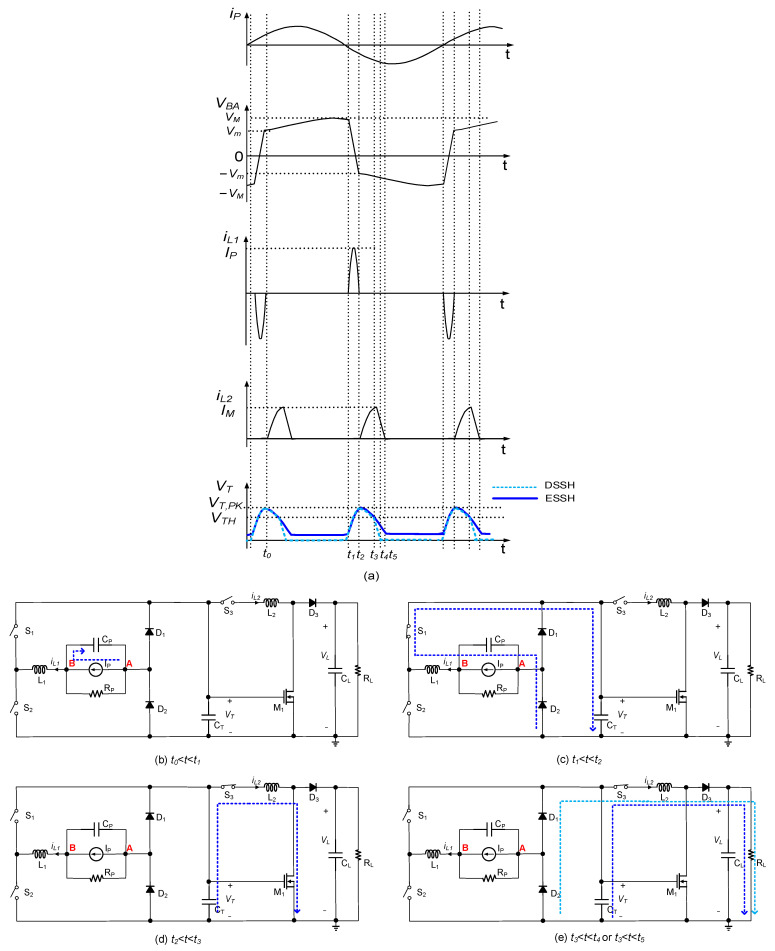
Configuration of the proposed circuit (**a**) voltage and current waveform (**b**) during *t*_0_ < *t* < *t*_1_, (**c**) during *t*_1_ < *t* < *t*_2_, (**d**) during *t*_2_ < *t* < *t*_3_, and (**e**) during *t*_3_ < *t* < *t*_4_ or *t*_3_ < *t* < *t*_5_.

**Figure 5 micromachines-14-01639-f005:**
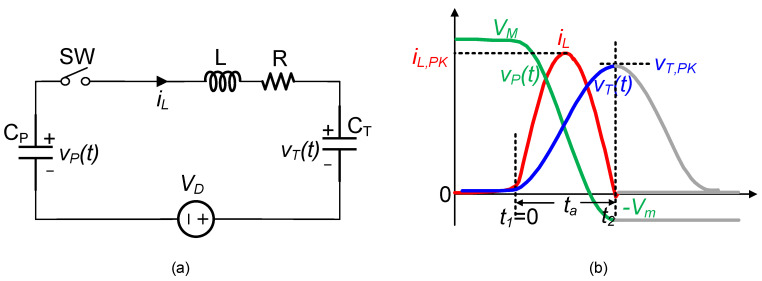
(**a**) Circuit model of the first resonant loop; (**b**) voltage and current waveforms.

**Figure 6 micromachines-14-01639-f006:**
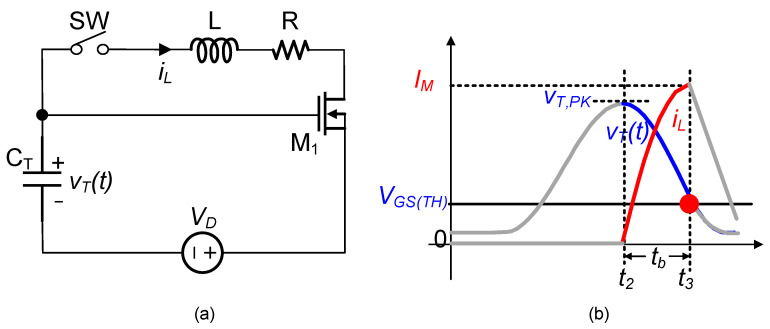
(**a**) Circuit model of the second resonant loop; (**b**) voltage and current waveforms.

**Figure 7 micromachines-14-01639-f007:**
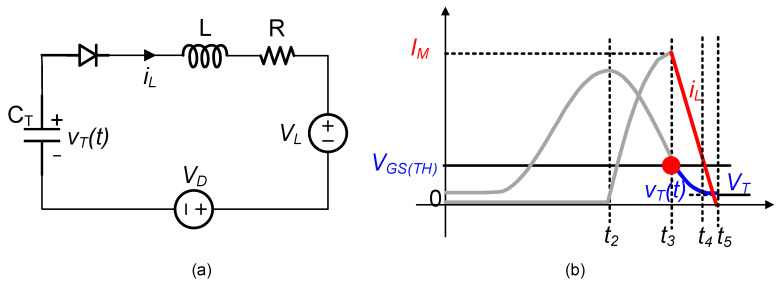
(**a**) Circuit model of the third resonant loop; (**b**) voltage and current waveforms.

**Figure 8 micromachines-14-01639-f008:**
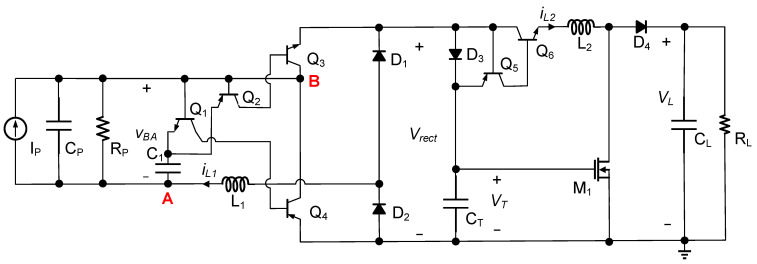
Proposed DSSH circuit with MOSFET threshold voltage management.

**Figure 9 micromachines-14-01639-f009:**
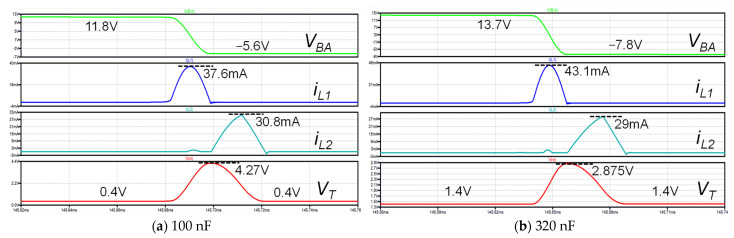
Simulation waveforms in the DSSH and ESSH schemes with different capacitor values.

**Figure 10 micromachines-14-01639-f010:**
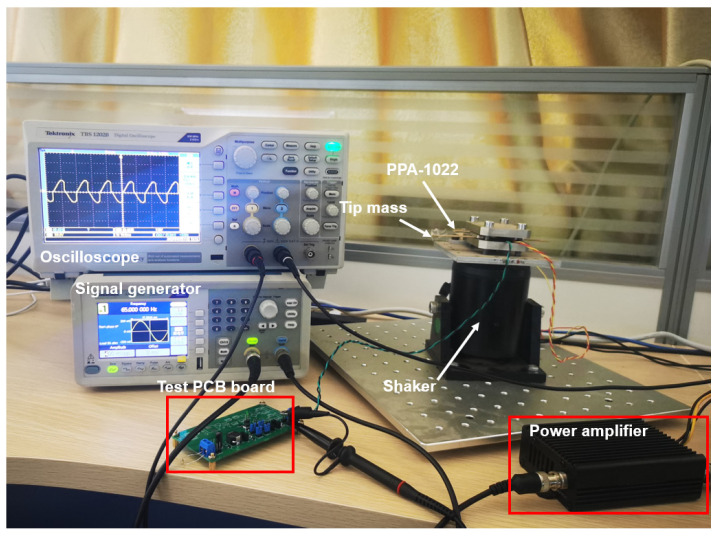
Experimental setup.

**Figure 11 micromachines-14-01639-f011:**
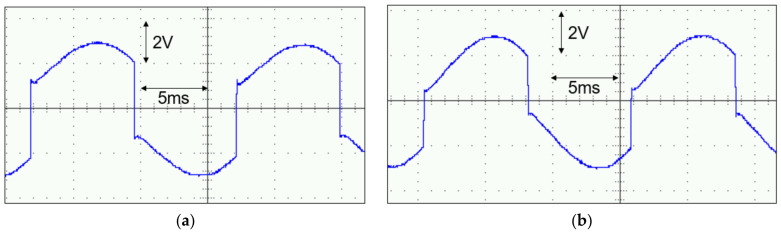
Measured waveforms of the PE cantilever voltage *V_BA_* (**a**) with rectifier only and (**b**) with the proposed circuit.

**Figure 12 micromachines-14-01639-f012:**
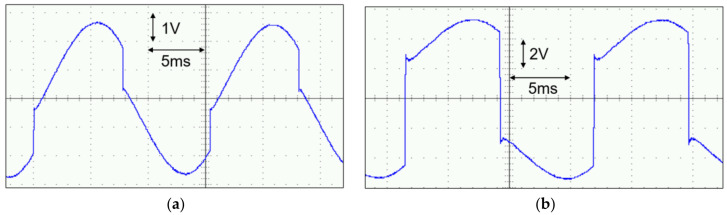
Measured waveforms of the PE cantilever voltage *V_BA_* with an intermediate capacitor of (**a**) 22 nF and (**b**) 320 nF.

**Figure 13 micromachines-14-01639-f013:**
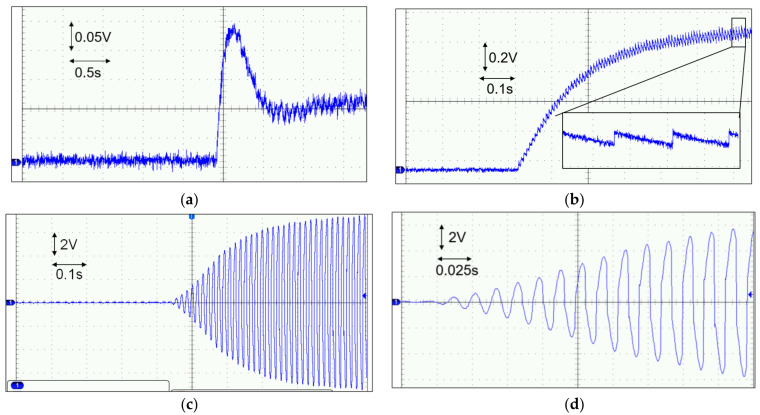
Cold start of the circuit measured voltage *V_L_* (**a**) when the cold start failed and (**b**) when the cold start succeeded. (**c**) The measured PE voltage *V_BA_*; (**d**) the zoomed-in voltage waveform.

**Figure 14 micromachines-14-01639-f014:**
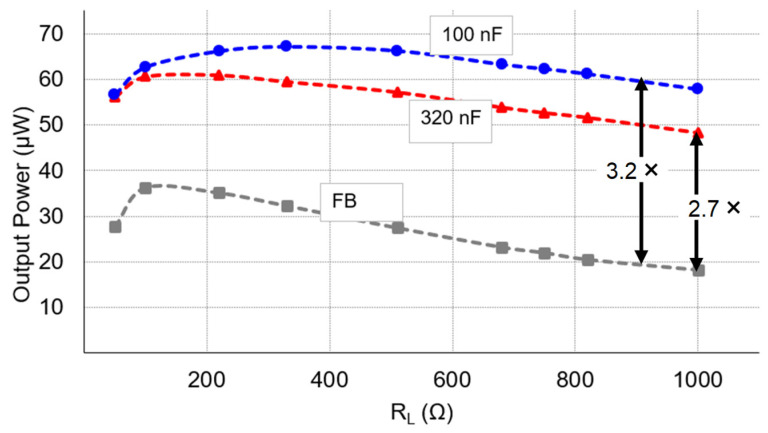
Power delivered to the load using two different intermediate capacitors with values of 100 nF and 320 nF.

**Figure 15 micromachines-14-01639-f015:**
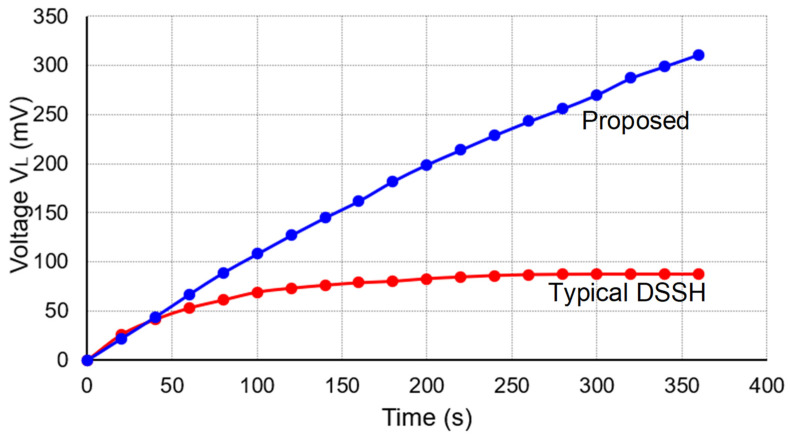
The output voltage *V_L_* versus the time for two different type circuits, the proposed circuit and the typical DSSH circuit, with a load of only a 10 mF capacitor.

**Table 1 micromachines-14-01639-t001:** The range of parameter *x* under two different schemes.

*x*	
DSSH	ESSH
$\begin{array}{l} \frac{1}{\lambda -\frac{\left(V_{E}+V_{D}\right)\left(1+\lambda \right)}{2V_{oc}}}<x\\ <\frac{2\left(V_{oc}-V_{E}-V_{D}\right)\left(1+\lambda \right)-2\left(V_{GS\left(TH\right)}\sqrt{1+\frac{1}{\eta }}-V_{E}\right)}{V_{GS\left(TH\right)}\sqrt{1+\frac{1}{\eta }}-V_{E}} \end{array}$	$\begin{array}{l} \frac{2\left(V_{oc}-V_{E}-V_{D}\right)\left(1+\lambda \right)-2\left(V_{GS\left(TH\right)}\sqrt{1+\frac{1}{\eta }}-V_{E}\right)}{V_{GS\left(TH\right)}\sqrt{1+\frac{1}{\eta }}-V_{E}}<x\\ <\frac{2\left(V_{oc}-V_{E}-V_{D}\right)\left(1+\lambda \right)-2\left(V_{GS\left(TH\right)}-V_{E}\right)}{V_{GS\left(TH\right)}-V_{E}} \end{array}$

**Table 2 micromachines-14-01639-t002:** Simulation parameters.

Components	Value
PE transducer current *i_P_*	70 μA
Frequency *f_P_*	100 Hz
Capacitor C_P_	22 nF
External inductor $L_{1}$	1.5 mH (4.2 Ω in series)
External inductor $L_{2}$	1 mH (1.253 Ω in series)
Load capacitor $C_{L}$	1 μF
Load resistor $R_{L}$	100 kΩ

**Table 3 micromachines-14-01639-t003:** Power dissipation breakdown in simulation.

Components	100 nF		320 nF	
	Percentage	Power (μW)	Percentage	Power (μW)
Transistors $Q_{3}$ and $Q_{4}$	37%	39	36%	46.4
Diodes $D_{1}$ and $D_{2}$	18%	18.7	19%	23.8
Diode $D_{3}$	17%	18.1	18%	23.2
Inductor $L_{1}$	8%	9	11%	13.4
Diode $D_{4}$	7%	7	5%	7
Transistor $Q_{5}$ and $Q_{6}$	6%	6.2	5%	6
Transistors $Q_{1}$ and $Q_{2}$	5%	5.1	4%	5.7
Inductor $L_{2}$	2%	2	2%	2.44

**Table 4 micromachines-14-01639-t004:** Comparison of recent PE energy harvesting circuits.

Publication	TUFFC 2008 [21]	SMS 2010 [22]	SAP 2021 [30]	Actuators 2021 [31]	This Work
Process Technology	Discrete Components	Discrete Components	Discrete Components	Discrete Components	Discrete Components
PE Transducer	Custom	Custom	Custom MEMS	Custom	Custom
Extraction Scheme	DSSH	ESSH	DR-DSSH	EDSSH	DSSH/ESSH
*C_P_* (nF)	30	84/168	1.6/16	62.36	22
*f_P_* (Hz)	105.3	31.72	23	1–312.5	65
*V_oc_* (V)	-	-	12	12.26	5.4
Self-powered	No	Yes	Yes	Yes	Yes
Difficult to Implement	Hard	Hard	Medium	Medium	Easy
Inductor Value (μH)	1 H	-	2:1 Transformer	1 mH, 1 mH	1.2 mH, 1 mH
Number of Components	11 *	44	28	27	16
Charging Efficiency (%)	-	184 **	213	151	510

* Does not include a controller. ** Calculated in this paper.

## Data Availability

No new data were created or analyzed in this study. Data sharing is not applicable to this paper.

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
