# Peer review of "A Self-Powered DSSH Circuit with MOSFET Threshold Voltage Management for Piezoelectric Energy Harvesting"

_micromachines, 2023, doi:10.3390/mi14081639_

Round 1
Reviewer 1 Report
This paper proposed a DSSH Circuit for PEHs, and the utilization of MOSFET can provide a Threshold Voltage Management for the circuit. The design, prototype and experiments were reported. However, the paper is not good enough for publication.
1.The introduction should be improved. The motivation, target and novelty cannot be shown by reading this section. Why do you propose this work, what are your targets and how can you achieve them?
2. The necessity of Section 2 is not clear. It is a background for this work, but is not necessary for this paper. Maybe, this section can be blend in the first one.
3. The circuit for PEHs is a basic and common issue, which has been widely studied. It is still necessary to propose a comparison between your work and the published ones to show the real value of your work, e.g. the efficiency, promoting ability and number of utilized components, etc. The comparison in the paper is far away from satisfaction.
4. The stimulating signal for shaker is not given. Have you tested your circuit under different vibrations?
The English writing is good, and improcing some minor errors can make it better.
Author Response
Thank you for your comemnts. Please see the attachment.

Reviewer 2 Report
Dear authors,
first many thanks for your contribution. The topic and application are quite interesting to the field. The proposed circuit could be used in some piezoelectric harvesting systems.
Some comments:
-some abbreviations need to be clarified: such, S-SSHI or MPPT.
-please recheck page 4, Figure 3. The current through SW1 is iL and not iL1. Please correct.
-All tables and figures have to be self-explaining. Please provide as much as possible information to the figures what is seen.
- It would be really valuable if authors can provide at the end a recommended general optimized design reules (with formulation) for a general DSSH circuit.
Sincerely,
Dear authors,
the text is quite good written. Easy to read. But still small errors. And important that all tables and figures have to be self-explaining. Please add more information. It helps readers to follow and understand quickly.
Sincerely,
Round 2
Reviewer 1 Report
The revission work is good, and I think the paper can be accepted now.
Here is a littile advice about the vibrationin the test. It is better to show the form of tested vibrations: e.g. sinusoidal, random or others.